# Temperature-Driven Divergence in Microbial Consortia and Physicochemical Functionality: A Comparative Study of High- and Medium-Temperature *Daqu*

**DOI:** 10.3390/microorganisms13061312

**Published:** 2025-06-05

**Authors:** Huawei Yuan, Jia Zheng, Liping Ding, Hong Wang, Qin Jiang, Chao Zhang, Tingna Xie, Guohui Nan, Li Li, Kai Lou

**Affiliations:** 1Solid-State Fermentation Resource Utilization Key Laboratory of Sichuan Province, Faculty of Quality Management and Inspection Quarantine, Sichuan Higher Education Engineering Research Center for Agri-Food Standardization and Inspection, Yibin University, Yibin 644000, China; yuanhuawei001@163.com (H.Y.); 19182431312@163.com (L.D.); jiangqin888666@163.com (Q.J.); chaozhang8760@163.com (C.Z.); xietingna@yibinu.edu.cn (T.X.); nanguohui@yibinu.edu.cn (G.N.); lilijc201109@163.com (L.L.); 2Yibin Wuliangye Co., Ltd., Yibin 644007, China; zhengwanqi86@163.com (J.Z.); wanghong@wuliangye.com.cn (H.W.)

**Keywords:** Chinese *Baijiu*, thermophilic bacteria, *Bacillus*, *Aspergillus*, metabolic trade-offs

## Abstract

*Daqu*, a crucial fermentation starter for Chinese *Baijiu*, develops distinct microbial and physicochemical profiles depending on fermentation temperature, which significantly influence enzymatic activity and flavor formation. While high-temperature (HT-*Daqu*, 65 °C) and medium-temperature (MT-*Daqu*, 60 °C) variants are known to produce different liquor aromas, systematic comparisons of their microbial and physicochemical dynamics remain limited. This study integrated physicochemical assays (moisture, starch, acidity, enzymatic activity) with 16S rRNA and ITS (Internal Transcribed Spacer) sequencing to analyze HT-*Daqu* (HQ1–HQ3) and MT-*Daqu* (MQ1–MQ3) from Sichuan breweries. Results revealed that HT-*Daqu* exhibited significantly lower moisture (*p* < 0.05) and starch content (*p* < 0.05) but higher acidity *(p* < 0.05) compared to MT-*Daqu*. Enzymatic activities were generally reduced in HT-*Daqu*, except for neutral protease. Microbial profiling revealed distinct microbial dynamics between HT-*Daqu* and MT-*Daqu*: HT-*Daqu* harbored thermophilic *Bacillus* (40–60% relative abundance) with reduced fungal diversity, while MT-*Daqu* prioritized fungal consortia—*Aspergillus* dominated MQ1 (78%) and *Saccharomyces* transiently peaked in MQ2 (35%)—which correlated with enhanced saccharification enzyme activities and esterification potential. Alpha-diversity indices confirmed higher bacterial diversity in HT-*Daqu* and greater fungal richness in MT-*Daqu*. Correlation networks highlighted temperature-driven linkages, such as *Bacillus* positively associating with acidity. These findings elucidate the trade-offs between microbial stress adaptation and metabolic efficiency under different thermal regimes, providing actionable insights for optimizing *Daqu* production through targeted microbial management and temperature control to enhance liquor quality.

## 1. Introduction

Chinese *Baijiu*, a traditional distilled spirit and one of the world’s oldest fermented beverages, is renowned for its complex flavor profiles and cultural significance. Its production relies on a starch-rich substrate (e.g., sorghum, wheat) that undergoes solid-state fermentation, a process driven by *Daqu*—a microbial-rich starter culture critical for saccharification, alcohol synthesis, and flavor development [1]. *Daqu* is crafted by fermenting crushed grains under open conditions, enabling natural inoculation of environmental microorganisms, including bacteria, molds, yeasts, and actinomycetes [2]. These microbes produce hydrolytic enzymes (e.g., amylases, proteases) and metabolic precursors that collectively shape the aroma, texture, and quality of *Baijiu* [1].

*Daqu* is categorized by its peak fermentation temperature: high-temperature *Daqu* (HT-*Daqu*; 60–65 °C), medium-temperature *Daqu* (MT-*Daqu*; 50–60 °C), and low-temperature *Daqu* (LT-*Daqu*; <50 °C) [2]. Each type supports distinct microbial consortia and physicochemical properties, which directly correlate with the flavor characteristics of the final product [3]. For instance, HT-*Daqu*, pivotal for sauce-aroma *Baijiu* (e.g., Moutai), harbors thermotolerant *Bacillus* and *Aspergillus* species that generate pyrazines and phenolic compounds under prolonged high-temperature fermentation [3]. MT-*Daqu*, used for strong-aroma liquors, exhibits balanced microbial diversity dominated by *Saccharopolyspora* and *Lactobacillus*, contributing to ester-rich profiles [3]. LT-*Daqu*, employed in light-aroma varieties, favors *Rhizopus* and *Pichia* species, which enhance fruity and floral notes through cooler, shorter fermentation cycles [3]. These temperature-driven differences in microbial composition also influence enzymatic activities, such as alcohol acyltransferases (AATs), which catalyze ester synthesis and are highly temperature-dependent [4].

Key physicochemical parameters—including moisture, starch content, acidity, and enzyme activities (e.g., saccharification, liquefaction)—are highly sensitive to thermal conditions. HT-*Daqu* typically exhibits reduced starch content and lower enzymatic activity (except neutral protease) compared to MT-*Daqu*, likely due to heat-induced inactivation [5]. Conversely, MT-*Daqu* fosters fungal dominance (e.g., *Aspergillus* and *Saccharomyces*), enhancing saccharification and ester synthesis [6]. These differences highlight temperature-regulated trade-offs between microbial stress adaptation and metabolic efficiency. However, controversies persist regarding the trade-offs between microbial diversity and functional enzyme activity under different thermal regimes, necessitating systematic comparisons [7].

Microbial community structure further differentiates *Daqu* functionality. HT-*Daqu* selects for thermophilic bacteria like *Bacillus*, which contribute to protein degradation and aroma precursor synthesis under extreme heat [8]. In contrast, MT-*Daqu* supports synergistic fungal–bacterial consortia (e.g., *Saccharomycopsis* and *Lactobacillus*) that optimize starch hydrolysis [9]. Recent advances in high-throughput sequencing confirm that HT-*Daqu* harbors greater bacterial diversity but reduced fungal richness compared to MT-*Daqu* [10]. This temperature-driven microbial succession underscores the need for comparative analyses to guide *Daqu* process optimization. Nevertheless, the mechanistic links between temperature-induced microbial shifts and key physicochemical parameters, such as neutral protease activity or reducing sugar content, remain underexplored [11].

Despite advances in understanding *Daqu*’s microbiome, critical knowledge gaps persist. Comparative studies on temperature-dependent microbial assembly patterns between HT- and MT-*Daqu* remain sparse, particularly regarding functional shifts in enzyme kinetics and metabolic networks. We hypothesize that *Daqu*-production temperature governs enzymatic profiles and microbial diversity, with HT-*Daqu* favoring bacterial dominance over fungal activity. By integrating conventional physicochemical assays (moisture, starch, acidity, reducing sugars) and enzyme activity measurements (saccharification, liquefaction, neutral protease) with high-throughput sequencing, we systematically evaluated microbial and functional divergences. The findings elucidate temperature-driven trade-offs in microbial functionality and offer actionable strategies to optimize fermentation for enhanced liquor quality.

## 2. Materials and Methods

### 2.1. Sample Collection

HT-*Daqu* (HQ1–HQ3 denote high-temperature *Daqu* replicates; peak fermentation temperature: 65 °C) and MT-*Daqu* (MQ1–MQ3 represent medium-temperature *Daqu* replicates; peak fermentation temperature: 60 °C) were obtained from three independent brewing enterprises in Yibin City and Luzhou City, respectively. The *Daqu* samples were produced using a standardized protocol: locally sourced soft wheat was first crushed, mixed with water (approximately 40–45% moisture content), and pressed into brick-shaped blocks (dimensions: 25 × 15 × 6 cm). These blocks underwent spontaneous solid-state fermentation under controlled temperature and humidity conditions. For HT-*Daqu*, the fermentation process included a high-temperature stage (peaking at 65 °C for 5–7 days) to promote thermophilic microbial succession (e.g., *Bacillus*, *Thermoascus*) and enzyme activation, while MT-*Daqu* maintained moderate thermal conditions (peaking at 60 °C for 3–5 days) to preserve fungal diversity (e.g., *Aspergillus*, *Saccharomyces*). Post-fermentation, the *Daqu* bricks were air-dried naturally for 15–20 days and stored in ventilated rooms at ambient temperature (20–25 °C) for 3–4 months to ensure microbial stabilization and flavor maturation. Prior to analysis, samples were collected from the outer, middle, and inner layers of the *Daqu* bricks to account for spatial heterogeneity, homogenized under sterile conditions, and stored at −80 °C for microbial profiling and 4 °C for enzymatic assays. At the time of sampling, the *Daqu* exhibited key physicochemical characteristics: moisture content (10.5–12.8%), acidity (pH 5.2–5.8), liquefying activity (1.2–1.6 U/g), and esterifying capacity (120–150 mg/100 g), which correlate with their enzymatic and microbial functionality during *Baijiu* fermentation.

Eighteen samples (9 per group) were collected in August 2024 at the end of the microbial incubation phase (45 days for HT-*Daqu*; 30 days for MT-*Daqu*) (Appendix A). To ensure sample homogeneity, each *Daqu* block was homogenized via a three-point sampling protocol (core, crust, and edge regions). Each homogenized sample (600 g) was divided into two aliquots: 300 g stored at −20 °C for physicochemical characterization, and 300 g snap-frozen in liquid nitrogen and preserved at −80 °C for microbial DNA extraction (16S rRNA gene and ITS sequencing). Biological triplicates (three independent fermentation batches per group) were included to enhance experimental robustness.

### 2.2. Physicochemical Properties and Enzyme Activity Analysis

The physicochemical properties of *Daqu*, including moisture, acidity, starch content, liquefying enzyme activity (measured via α-amylase-mediated hydrolysis of starch into dextrins at 65 °C), saccharifying enzyme activity (assessed via glucoamylase-driven conversion of dextrins to glucose at 60 °C), fermentation capacity (quantified by CO_2_ production during 72-h anaerobic fermentation of *Daqu* slurry, normalized to glucose equivalents), and esterification capacity, were measured according to the Brewing *Daqu* Quality Control Standards (QB/T 4257-2011) [12] (Appendix A). Protease activities (acidic and neutral) were evaluated using the Technical Specifications for Strong-Flavor *Daqu* Analysis (DB 34/T 3085-2018) [13]. Cellulase activity was determined via the spectrophotometric protocol outlined in the Feed Additive Testing Guidelines (NY/T 912-2020) [14], while hemicellulase activity was analyzed following the Xylanase Assay Method for Feed Additives (GB/T 23874-2009) [15]. Reducing sugar content was quantified through the National Food Safety Standard (GB 5009.7-2016, Method 2) [16], with values reported as glucose equivalent concentrations.

### 2.3. Sample DNA Extraction and Microbial Community Structure Analysis

Total genomic DNA was extracted from 0.3 g of wheat-based *Daqu* samples using the PowerSoil^®^ DNA Isolation Kit (MO BIO Laboratories, Carlsbad, CA, USA) according to the manufacturer’s instructions, with triplicate extractions per sample pooled to minimize variability. DNA concentration and purity were evaluated by spectrophotometry (NanoDrop, Wilmington, DE, USA) and agarose gel electrophoresis. The V3-V4 regions of bacterial 16S rRNA genes were amplified using primers 338F (5′-ACTCCTACGGGAGGCAGCAG-3′) and 806R (5′-GGACTACHVGGGTWTCTAAT-3′) [17], while fungal ITS regions were amplified with primers ITS1F (5′-CTTGGTCATTTAGAGGAAGTAA-3′) and ITS4 (5′-TCCTCCGCTTATTGATATGC-3′) [18]. Purified amplicons were sequenced on the Illumina MiSeq platform (2 × 250 bp paired-end) at Biomarker Technologies Co., Ltd. (Beijing, China), with raw data processed through QIIME2 (v2021.11) for quality filtering, denoising, and feature table generation.

### 2.4. Bioinformatics Analysis

Raw sequencing data were analyzed using CCS 6.0.0 (PacBio, Menlo Park, CA, USA) to produce high-accuracy circular consensus reads. Quality filtering involved adapter removal through the fastx-toolkit_v0.0.14-6 and elimination of chimeric sequences with VSEARCH v2.14.1’s reference-based uchime method. Sequence deduplication was executed via the fastx_uniques function in USEARCH v11.0.667, followed by Amplicon Sequence Variant (ASV) derivation through the unoise3 pipeline (minimum read abundance: minsize = 8). Taxon-specific filtration was implemented using in-house scripts to retain fungal sequences from ITS regions and bacterial sequences from 16S datasets (Appendix A). Operational taxonomic units (OTUs) were defined through 97% similarity clustering with USEARCH (v9.2.612), followed by taxonomic classification via NCBI NT database alignment using BLASTn v2.2.28 (e-value cutoff: 1 × 10^−5^, minimum similarity: 97%). α-diversity assessments were performed in R v4.3.0 using the vegan package [19], with calculations based on normalized ASV abundance matrices.

### 2.5. Alpha Diversity Analysis

To systematically characterize microbial community profiles, three α-diversity metrics (Chao1, Shannon, Simpson) were utilized. The Chao1 index quantifies taxonomic abundance, whereas Shannon and Simpson indices incorporate both population quantity and distribution uniformity—essential parameters for evaluating metabolic consistency and functional adaptability within *Daqu* fermentation systems. Elevated Shannon indices coupled with reduced Simpson coefficients correlate with enhanced biodiversity, suggesting an equilibrated community structure. All statistical processing was performed within the QIIME2 computational framework.

### 2.6. Statistical Analysis

The data were analyzed using SPSS 26.0 (IBM Corp., Armonk, NY, USA). Differences among groups were evaluated by one-way analysis of variance (ANOVA), followed by Tukey’s HSD post hoc test, with significance defined at *p* < 0.05. Spearman correlation analysis, conducted using SAS v8 (SAS Institute Inc., Cary, NC, USA), was employed to examine the relationships between dominant Daqu microbial taxa and physicochemical parameters. The results were visualized using Cytoscape v3.7.2 (Cytoscape Consortium, San Diego, CA, USA).

## 3. Results

### 3.1. Daqu-Making Temperature Modulates Physicochemical Properties

All *Daqu* samples exhibited water content below 13% (*w*/*w*) (Figure 1A), conforming to standard fermentation parameters. HT-*Daqu* (HQ1–3) demonstrated significantly lower moisture (8.88–10.75%) than MT- counterparts (MQ1–3: 11.68–12.51%) (*p* < 0.05) yet displayed elevated acidity (1.70–2.16 mmol/10 g vs. 1.10–1.22 mmol/10 g; Figure 1A; Appendix A). Intra-group variations revealed extremes in water content (MQ1: 12.51%; HQ1: 8.88%) and acidity (HQ2: 2.16 mmol/10 g; MQ3: 1.09 mmol/10 g).

HT-*Daqu* (HQ1–3) exhibited significantly lower reducing sugar (0.75–0.94%) and starch content (54.04–57.93%) compared to MT-*Daqu* (MQ1–3: 1.68–1.86% reducing sugar, 59.50–61.20% starch) (*p* < 0.05, Figure 1B; Appendix A). Notable intra-group variations were observed, with HQ3 (0.75%) and MQ2 (1.86%) representing the respective extremes of reducing sugar content, while starch content ranged from 54.04% (HQ2) to 61.20% (MQ2).

### 3.2. Temperature-Driven Regulation of Enzymatic Activities

Significant temperature-dependent variations in enzymatic profiles were observed between MT and HT *Daqu*. MT-*Daqu* exhibited 1.6-fold higher neutral protease activity than HT-*Daqu* (*p* < 0.05; Figure 2A; Appendix A), whereas acidic protease activity was markedly reduced in MT-*Daqu*. Saccharification enzyme activity in MT-*Daqu* (MQ2) surpassed that of HT-*Daqu* (HQ1) by 38% (*p* < 0.05), though comparable liquefaction activity was maintained across all groups (Figure 2B). Cellulase and hemicellulase activities peaked in MT-*Daqu* (MQ1), showing 2.3- and 1.9-fold increases relative to HT-*Daqu* (*p* < 0.01; Figure 2C; Appendix A). Notably, MT-*Daqu* demonstrated superior esterase activity and fermentation capacity, with MQ1 outperforming all HT-*Daqu* samples by 27% (*p* < 0.05; Figure 2D).

### 3.3. Daqu-Making Temperature Shapes α-Diversity

Alpha-diversity metrics of bacterial communities revealed systematic differences between HT-*Daqu* (HQ1–3) and MT-*Daqu* (MQ1–3). HT-*Daqu* exhibited significantly higher ASV counts (HQ1: 961 vs. MQ3: 77; *p* < 0.01; Table 1), while MT-*Daqu* MQ1 showed the highest Chao1 index (642.03 ± 0.021) (Table 1), surpassing all HT counterparts (e.g., HQ1: 396.36 ± 2.810). The Shannon index was significantly elevated in HT-*Daqu* (HQ1: 7.83 ± 0.073 vs. MQ3: 1.76 ± 0.042), whereas the Simpson index displayed an inverse pattern (HQ1: 0.039 ± 0.001 vs. MQ1: 0.980 ± 0.002, Table 1). No significant difference was observed in the Shannon index between HQ2 (4.81 ± 0.009) and MQ1 (4.35 ± 0.008) (*p* > 0.05). All samples achieved sequencing coverage > 99.9% (Table 1), with HT-*Daqu* demonstrating greater intra-group variability in Chao1 indices (e.g., HQ1: ±2.810) compared to MT groups (e.g., MQ1: ±0.021).

Alpha diversity analysis of fungal communities in HT (HQ1–3) and MT-*Daqu* (MQ1–3) revealed significant ecological divergence (Table 2). MT-*Daqu* exhibited markedly higher species richness, with ASV counts (635–2037) and Chao1 indices (209.79–952.11) substantially exceeding those of HT groups (ASVs: 331–368; Chao1: 47.83–122.24; Table 2). The maximum Chao1 value was recorded in MQ2 (952.11 ± 0.004), contrasting sharply with the minimum in HQ2 (47.83 ± 5.137), demonstrating statistically significant intergroup variation (*p* < 0.05). Species diversity patterns further distinguished the groups. MT- *Daqu* showed significantly higher Shannon indices (MQ1: 8.44 ± 0.022; MQ2: 7.63 ± 0.010) compared to HT counterparts (1.53–2.01). Conversely, Simpson indices revealed inverse trends (Table 2), with HT groups displaying lower dominance values (0.37–0.63) than MT samples (0.64–0.99). Sequencing coverage exceeded 99.9% for all samples (Table 2), ensuring robust community characterization through sufficient sequencing depth. These metrics collectively highlight temperature-dependent restructuring of fungal community complexity in *Daqu* ecosystems.

### 3.4. Genus-Level Abundance Dynamics Driven by Qu-Making Temperature

Bacterial profiling identified 856 genera across HT- and MT-*Daqu*. HT variants were dominated by *Bacillus* (40–60% relative abundance), *Kroppenstedtia,* and *Caldibacillus* (Figure 3). In contrast, MT-*Daqu* exhibited predominance of mesophilic genera including *Virgibacillus* (particularly in MQ1), *Saccharopolyspora*, and *Desmospora* (Figure 3). Minor taxa (unclassified genera) constituted <20% of communities in both groups.

Fungal community analysis identified 932 genera across HT- and MT-*Daqu*. HT- *Daqu* exhibited dominance of thermotolerant taxa: *Paecilomyces* (52% relative abundance), *Talaromyces* (28%), and *Rasamsonia* (33%) (Figure 4). In contrast, MT-*Daqu* was characterized by mesophilic genera *Aspergillus* (78%), *Saccharomyces* (35%), and *Wickerhamomyces* (31%), demonstrating temperature-driven fungal consortia stratification (Figure 4).

### 3.5. Microbial-Physicochemical Correlations in HT- and MT-Daqu

The study revealed significant correlations between microbial taxa and physicochemical properties in HT- and MT-*Daqu*. Bacterial genera exhibited distinct functional roles: *Sphingomonas* showed highly significant positive associations with acid protease and glucoamylase activities (*p* < 0.01) (Figure 5), alongside significant correlations with esterase activity and fermentation capacity (*p* < 0.05), but displayed a strong negative correlation with acidity (*p* < 0.01). *Bacillus* was positively linked to acidity but negatively correlated with moisture content (*p* < 0.01) and reducing sugar/esterase activity (*p* < 0.05). *Desmospora* demonstrated significant positive correlations with amylase and hemicellulase activities (*p* < 0.01), while *Scopulibacillus* was negatively associated with starch content and neutral protease activity (*p* < 0.05) (Figure 5). *Kroppenstedtia* exhibited dual correlations—positive with acidity and negative with acid protease, glucoamylase, and fermentation capacity.

Fungal genera displayed contrasting behaviors: *Aspergillus* negatively correlated with acid protease, glucoamylase, and esterase activities, diverging from its typical starch-degrading roles. *Cladosporium* and *Saccharomyces* showed significant positive associations with neutral protease (*p* < 0.05) and acid protease (*p* < 0.05) (Figure 6), respectively, while *Talaromyces* strongly negatively correlated with total sugar (*p* < 0.01). These findings highlight the temperature-dependent microbial–physicochemical interplay, emphasizing the multifunctionality of understudied taxa like *Sphingomonas* and the context-dependent roles of ubiquitous genera such as *Aspergillus*. The results provide actionable insights for optimizing fermentation parameters and microbial management in *Daqu* production.

## 4. Discussion

The findings of this study reveal critical temperature-dependent differences in *Daqu’s* physicochemical properties, offering both alignment and divergence from prior research. Consistent with Xing et al. [20], intra-group variations in moisture and acidity were observed, but our data uniquely demonstrated that high-temperature *Daqu* (HQ) exhibited significantly lower moisture (8.88–10.75%) and higher acidity (1.70–2.15 mmol/10 g) compared to medium-temperature *Daqu* (MQ: 11.68–12.51% moisture, 1.10–1.22 mmol/10 g acidity). This contrasts with Chen et al. [21], who reported higher acidity in medium-temperature *Daqu* dominated by *Weissella* and *Lactobacillus*. The elevated acidity in HQ may instead arise from thermotolerant acid-producing microbes (e.g., *Thermoascus* or *Bacillus* [22]), which thrive under high-temperature conditions and accelerate organic acid synthesis. Similarly, HQ’s lower reducing sugar (0.75–0.94%) and starch (54.04–57.93%) compared to MQ (1.68–1.86% reducing sugar; 59.50–61.20% starch) align with Liu et al. [23], who linked starch depletion to microbial activity. High temperatures likely enhance amylolytic enzyme activity (e.g., from *Rhizomucor* or *Aspergillus* [21]), accelerating starch hydrolysis. However, the rapid microbial consumption of reducing sugars in HQ for energy or secondary metabolites—less pronounced in MQ due to moderate enzymatic kinetics—explains its lower residual sugar levels.

The observed temperature-driven variations in enzymatic activities align with prior findings on microbial community dynamics and functional shifts in *Daqu*. Similar to Chen et al. [21], MT-*Daqu* exhibited higher saccharification and esterification capacities than HT-*Daqu*. While prior studies attribute such functional advantages to thermotolerant fungi (e.g., *Aspergillus*) and bacteria (e.g., *Lactobacillus*) under moderate temperatures [3,11], our data reveal a distinct pattern: although *Aspergillus* dominated fungal communities in the initial MT-*Daqu* batch (MQ1: 78%), its abundance sharply declined in subsequent batches (MQ2–3: <35%). Furthermore, *Lactobacillus* was undetected across all samples, whereas thermotolerant taxa such as *Bacillus* (22–36%) and *Virgibacillus* (18–24%) were enriched, suggesting temperature-driven selection of these genera likely underpins MT-*Daqu*’s metabolic superiority through enhanced carbohydrate metabolism and ester precursor synthesis.

The elevated neutral protease activity in MT-*Daqu* contrasts with HT-*Daqu’s* reduced acidic protease activity, possibly reflecting microbial adaptations to temperature-regulated pH and substrate availability [3]. Notably, the 2.3-fold higher cellulase activity in MT-*Daqu* parallels findings by Huang et al. [11], who linked enhanced lignocellulose degradation to *Bacillus*-dominated communities under moderate thermal stress. However, the comparable liquefaction activity across all groups diverges from Feng et al. [3], where HT-*Daqu* showed reduced amylase activity, suggesting strain-specific thermostability or compensatory metabolic pathways in this study. The superior esterase activity in MT-*Daqu* may stem from thermotolerant taxa such as *Virgibacillus* and *Bacillus*, which thrive under moderate thermal conditions (45–55 °C) and exhibit enzymatic synergy with fungi (e.g., *Aspergillus*) to drive ester precursor synthesis [24,25]. These results underscore temperature as a pivotal regulator of microbial consortia and their enzymatic repertoires, with MT-*Daqu’s* balanced microbial network enabling multifunctional enzymatic advantages.

The study revealed temperature-driven divergence in microbial α-diversity between HT and MT-*Daqu*. HT-*Daqu* exhibited higher bacterial ASV counts (HQ1: 961 vs. MQ3: 77) and Shannon indices (7.83 vs. 1.76), aligning with the dominance of thermotolerant *Bacillus* and *Thermoactinomyces* [11]. However, MT-*Daqu* unexpectedly surpassed HT-*Daqu* in bacterial Chao1 richness (MQ1: 642.03 vs. HQ1: 396.36), contrasting with Fu et al. [26], likely due to transient dominance of mesophilic taxa (*Lactobacillus*, *Weissella*) during early fermentation [23]. Fungal diversity diverged sharply: MT-*Daqu* showed superior richness (ASVs: 2037 vs. 331–368) and Shannon indices (8.44 vs. 1.53–2.01), consistent with moderate-temperature support for *Aspergillus* and *Thermoascus* activity [27]. In contrast, HT-*Daqu’s* lower fungal Simpson indices (0.37–0.63 vs. 0.64–0.99) deviated from Pang et al. [22], potentially reflecting regional humidity variations favoring extremophiles (*Thermoascus*) and over generalists (*Candida*) [6,28]. Thermal gradients in HT-*Daqu* likely drove bacterial niche partitioning between thermophiles and mesophiles [29], while MT-*Daqu’s* fungal versatility stemmed from *Aspergillus*-mediated polysaccharide degradation [3]. The elevated Chao1 in MT-*Daqu* may arise from rare taxa (*Pediococcus*) proliferating during cooling phases under reduced competition. These findings underscore temperature as a pivotal regulator of microbial assembly, highlighting the need for strain-level functional analyses to resolve thermal adaptation trade-offs [30], thereby guiding fermentation optimization for balanced diversity and functional stability.

The study revealed distinct microbial stratification between HT and MT-*Daqu*. Temperature-driven microbial consortia stratification in MT-*Daqu* involves *Virgibacillus* and *Saccharopolyspora* alongside *Aspergillus,* forming a multi-taxa functional network rather than single-genus dominance. This contrasts with studies reporting *Weissella* and *Lactobacillus* in other regional *Daqu* systems [31], suggesting temperature-driven microbial selection varies significantly across production lineages. However, the prominence of *Kroppenstedtia* in HT-*Daqu* contrasts with Deng et al. [32], potentially reflecting regional fermentation practices favoring its heat-resistant enzymes [33]. Fungal communities exhibited temperature-driven divergence: *Paecilomyces* (52%) and *Talaromyces* (28%) dominated HT-*Daqu*, aligning with thermotolerant taxa in Fu et al. [26] but differing from studies emphasizing *Thermoascus* [32,33], possibly due to *Paecilomyces’* superior amylolytic thermostability [25]. Unexpectedly low *Thermoascus* abundance in HT-*Daqu* diverged from Deng et al. [33], possibly reflecting transient temperature drops favoring *Paecilomyces* over sustained bio-heat (>50 °C) that enriches *Thermoascus* [6]. Thermophilic adaptations explain these patterns: *Bacillus* and *Kroppenstedtia* produce heat-stable proteases/amylases [11], while *Paecilomyces* outcompetes via sporulation efficiency [25]. Our findings on *Bacillus* and *Virgibacillus* dominance in MT-*Daqu* align with Zheng et al.’s categorization of *Daqu* types based on thermal regimes [34]. While Zheng et al. emphasized the prevalence of *Aspergillus* and *Rhizopus* in HT-*Daqu* [34], our data reveal a contrasting pattern in MT-*Daqu*, where *Bacillus* (22–36%) and *Virgibacillus* (18–24%) dominate after initial fungal decline. This divergence likely reflects temperature-driven microbial selection: MT-*Daqu*’s moderate thermal conditions (45–55 °C) favor thermotolerant *Bacillus* strains over mesophilic fungi like *Aspergillus*, corroborating recent studies on temperature-dependent microbial succession.

Our investigation of medium-high temperature *Daqu* microbiota-physicochemistry interplay demonstrates partial alignment with previous findings while revealing novel distinctions. Consistent with thermophilic biomarker identification [35], genus-specific enzymatic associations were observed, notably *Sphingomonas* showing strong positive correlations with acid protease (r = 0.72, *p* < 0.01) and glucoamylase (r = 0.68, *p* < 0.01), paralleling *Saccharomycopsis’* saccharification functions [36]. However, our data diverge from aroma-type *Daqu* studies [36,37], identifying *Sphingomonas* as a multifunctional taxon governing both starch hydrolysis and esterification pathways, potentially attributable to regional microbial biogeography [38]. *Bacillus* demonstrated significant acid tolerance (r = 0.54, *p* < 0.05), contrasting with *Lactobacillus*-dominated pH modulation in premium *Daqu* [37], suggesting metabolic niche specialization under thermal stress. Fungal dynamics revealed unexpected functional shifts: *Aspergillus* exhibited negative enzymatic correlations despite its established starch-degrading role [35], possibly due to thermophilic bacterial competition (e.g., *Thermoascus*), while *Cladosporium* maintained protease synergy as reported in barley-based systems [39]. Methodologically, we expanded microbial networks to encompass previously overlooked genera (e.g., *Desmospora*), contrasting with regional bacterial disparity studies [38]. The absence of fungal-esterification linkages contrasts with succession models [37], highlighting temperature–regime specificity. These findings advance *Daqu* microbial ecology by: (1) resolving functional redundancy/antagonism among ubiquitous taxa (e.g., *Kroppenstedtia’s* dual acid–enzyme correlations vs. *Thermoactinomyces* [40]); (2) identifying context-dependent enzymatic drivers (*Sphingomonas* vs. *Pantoea*/*Furfurilactobacillus* [36]; and (3) establishing temperature-mediated microbial trade-offs.

This study has certain limitations that warrant future investigation. While microbial diversity and enzymatic activities were characterized, the direct metabolic roles of key taxa (e.g., *Bacillus*, *Aspergillus*) in flavor formation remain unvalidated due to the absence of functional assays or multi-omics approaches. Additionally, enzymatic profiles were not correlated with volatile flavor compounds, leaving mechanistic connections between microbial activity and liquor aroma unresolved. Future research should prioritize integrating metagenomics, metatranscriptomics, and metabolomics to resolve functional contributions of microbial taxa to enzymatic and flavor pathways. Linking enzyme activities to volatile organic compounds via GC-MS would clarify direct flavor contribution mechanisms. Furthermore, employing advanced sequencing techniques (e.g., PacBio) or strain isolation could characterize thermophilic *Bacillus* subspecies and other biotechnologically relevant strains, enabling targeted microbial management to optimize Daqu functionality and liquor quality.

## 5. Conclusions

This study elucidates the pivotal role of fermentation temperature in governing microbial consortia assembly and functional divergence in *Daqu*, a key microbial starter for Chinese *Baijiu*. Comparative analysis of HT and MT-*Daqu* revealed temperature-driven ecological trade-offs: HT-*Daqu* exhibited bacterial dominance (*Bacillus* spp.) with enhanced thermotolerant adaptations, while MT-*Daqu* fostered fungal-rich communities (*Aspergillus*, *Saccharomyces*) demonstrating superior enzymatic capacities in carbohydrate metabolism and flavor precursor synthesis. These microbial divergences were mechanistically linked to temperature-modulated physicochemical parameters, including moisture, acidity, and starch utilization efficiency.

The findings from this study hold significant practical implications for the agri-food sector, particularly in optimizing traditional fermentation practices and enhancing product quality. The enzymatic superiority of MT-*Daqu* in fungal-driven ester synthesis (e.g., ethyl acetate, ethyl hexanoate) positions it as a strategic tool for *Baijiu* producers aiming to amplify fruity and floral flavor profiles, while HT-*Daqu*’s *Bacillus*-mediated proteolysis can be leveraged to elevate savory pyrazines (e.g., tetramethylpyrazine), aligning with market demands for Maotai-flavor *Baijiu*. By modulating fermentation temperatures (60 °C vs. 65 °C), producers can selectively enrich microbial consortia—prioritizing MT-*Daqu* for carbohydrate-active enzymes (amylases, hemicellulases) or HT-*Daqu* for thermostable proteases—thereby tailoring *Daqu* production to regional flavor preferences. Collectively, these insights bridge microbial ecology with biotechnology, providing a framework to harness microbial diversity for reproducible, high-quality fermentations while preserving the artisanal heritage of *Baijiu* production.

## Figures and Tables

**Figure 1 microorganisms-13-01312-f001:**
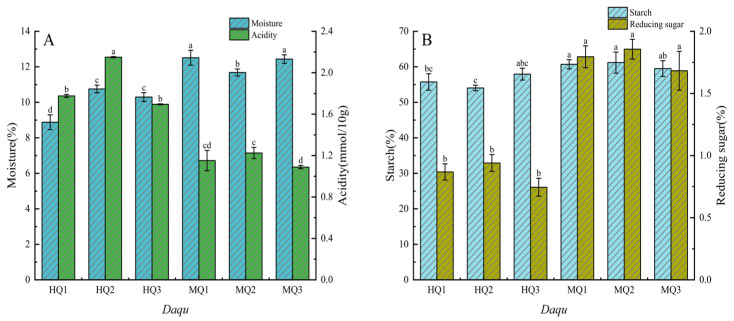
Changes in physicochemical indexes in different medium-high temperature *Daqu*. (**A**) Acidity, moisture. (**B**) Starch, reducing sugar. Data are presented as mean ± SD (*n* = 3). Bars with different letters indicate significant difference (*p* < 0.05).

**Figure 2 microorganisms-13-01312-f002:**
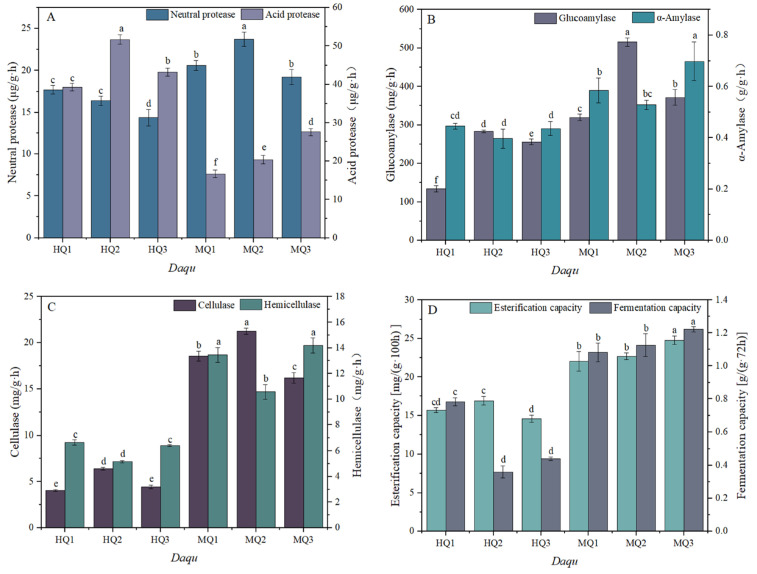
Changes in enzyme activity in medium-high temperature *Daqu*. (**A**) Neutral protease activity, acid protease. (**B**) Glucoamylase, α-amylase. (**C**) Cellulase, hemicellulase. (**D**) Fermentation capacity, esterification capacity. Data are presented as mean ± SD (*n* = 3). Bars with different letters indicate significant difference (*p* < 0.05).

**Figure 3 microorganisms-13-01312-f003:**
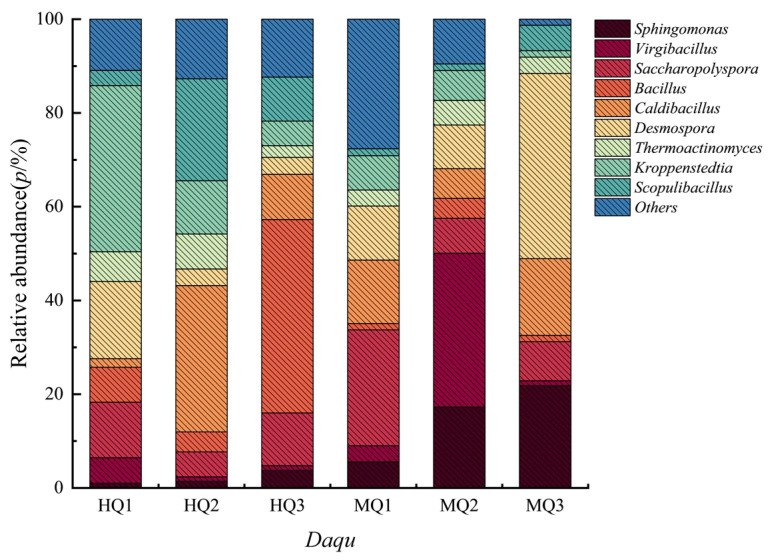
The bacterial taxonomic abundance in HT- and MT-*Daqu*.

**Figure 4 microorganisms-13-01312-f004:**
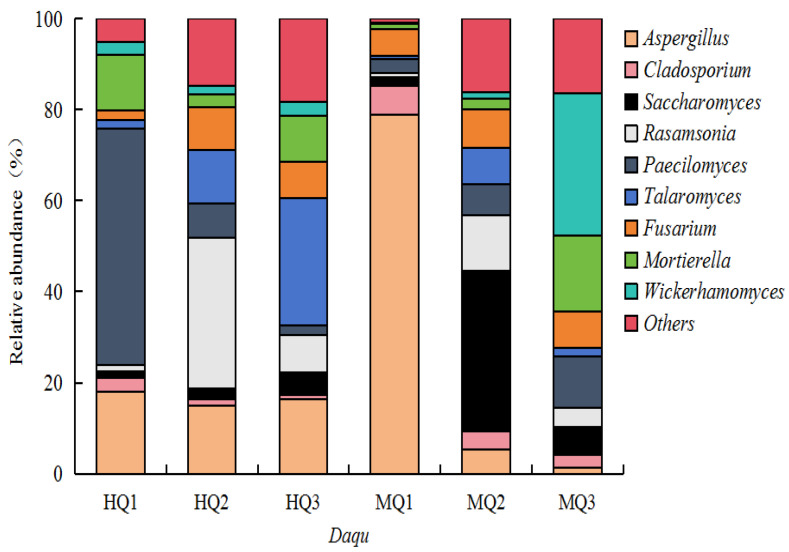
The fungal taxonomic abundance in HT- and MT-*Daqu*.

**Figure 5 microorganisms-13-01312-f005:**
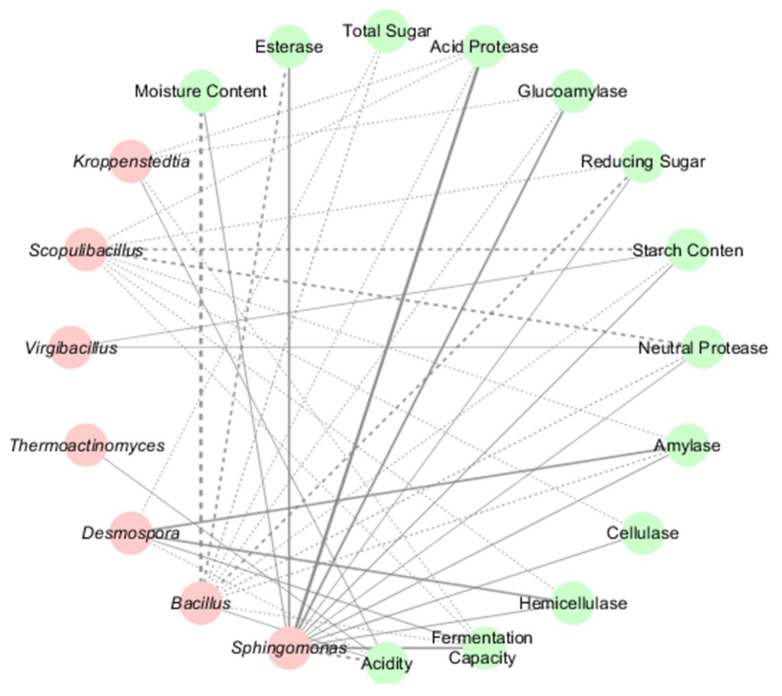
Bacterial–Physicochemical Network in HT- and MT-*Daqu*. Red spheres represent bacterial genera, while green spheres denote physicochemical indicators. Positive and negative associations are denoted by solid and dashed lines, respectively. Line color intensity corresponds to correlation strength (darker hues indicate higher absolute values), while bold lines signify statistically significant relationships (*p* < 0.05).

**Figure 6 microorganisms-13-01312-f006:**
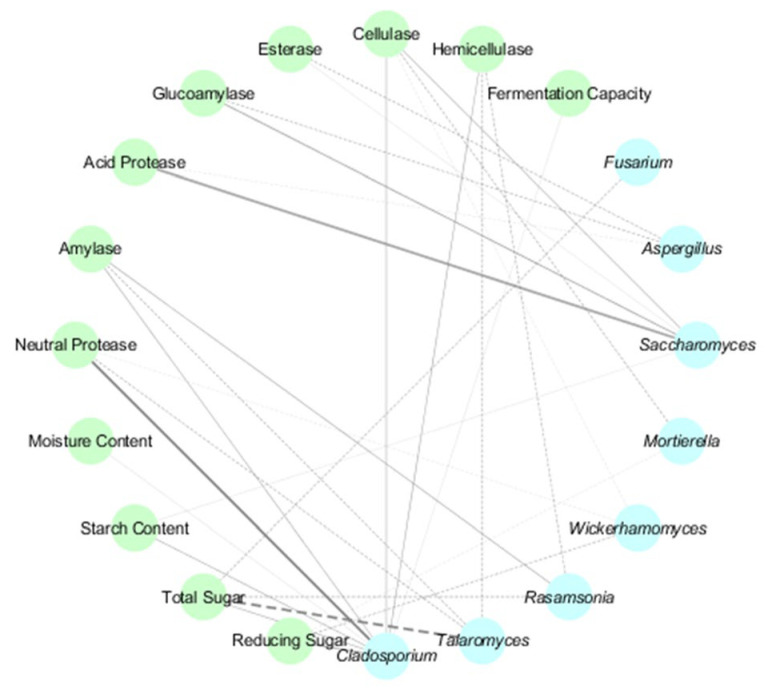
Fungal–Physicochemical Network in HT- and MT-*Daqu*. Blue spheres represent fungal genera, while green spheres denote physicochemical indicators. Positive and negative associations are denoted by solid and dashed lines, respectively. Line color intensity corresponds to correlation strength (darker hues indicate higher absolute values), while bold lines signify statistically significant relationships (*p* < 0.05).

**Table 1 microorganisms-13-01312-t001:** The impact of peak temperature on the α-diversity of bacterial communities in medium-high *Daqu*.

Samples	ASV Count	Chao1 Index	Shannon Index	Simpson Index	Coverage (%)
HQ1	961	396.36 ± 2.81 ^c^	7.83 ± 0.07 ^a^	0.04 ± 0.00 ^c^	99.99
HQ2	736	427.51 ± 0.17 ^f^	4.81 ± 0.01 ^c^	0.03 ± 0.00 ^d^	99.97
HQ3	795	242.61 ± 0.30 ^e^	5.64 ± 0.03 ^b^	0.04 ± 0.00 ^c^	99.97
MQ1	642	642.03 ± 0.02 ^a^	4.35 ± 0.01 ^d^	0.98 ± 0.00 ^a^	99.97
MQ2	322	322.07 ± 0.04 ^d^	2.17 ± 0.05 ^e^	0.92 ± 0.00 ^b^	99.96
MQ3	77	577.12 ± 0.01 ^b^	1.76 ± 0.04 ^f^	0.94 ± 0.00 ^ab^	99.94

Note: values with distinct superscript letters within the same row were statistically significant (*p* < 0.05).

**Table 2 microorganisms-13-01312-t002:** The impact of peak temperature on the α-diversity of fungal communities in medium-high temperature *Daqu*.

Samples	ASV Count	Chao1 Index	Shannon Index	Simpson Index	Coverage (%)
HQ1	368	122.24 ± 3.73 ^d^	2.01 ± 0.01 ^d^	0.37 ± 0.01 ^d^	99.93
HQ2	297	47.83 ± 5.14 ^f^	1.53 ± 0.05 ^e^	0.63 ± 0.00 ^b^	99.91
HQ3	331	49.03 ± 2.22 ^e^	1.91 ± 0.04 ^e^	0.51 ± 0.03 ^c^	99.93
MQ1	2037	209.79 ± 0.00 ^c^	8.44 ± 0.02 ^a^	0.99 ± 0.00 ^a^	99.99
MQ2	948	952.11 ± 0.00 ^a^	7.63 ± 0.01 ^b^	0.98 ± 0.00 ^a^	99.97
MQ3	635	636.34 ± 0.00 ^b^	3.72 ± 0.00 ^c^	0.64 ± 0.00 ^b^	99.96

Note: values with distinct superscript letters within the same row were statistically significant (*p* < 0.05).

## Data Availability

The data that support the findings of this study are available from the corresponding author upon reasonable request for non-commercial research purposes.

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
