# Peer review of "Temperature-Driven Divergence in Microbial Consortia and Physicochemical Functionality: A Comparative Study of High- and Medium-Temperature Daqu"

_microorganisms, 2025, doi:10.3390/microorganisms13061312_

Round 1
Reviewer 1 Report
Comments and Suggestions for Authors
Thank you for the opportunity to participate in the review of the manuscript entitled "Temperature-Driven Divergence in Microbial Consortia and Physicochemical Functionality: A Comparative Study of High- and Medium-Temperature Daqu".
The manuscript tells about medium- and high-temperature Daqu and the microbial diversity associated with them.
The manuscript has a typical layout for a research paper. The abstract is written correctly, it contains all the necessary elements. The introduction is written correctly. The cited literature is appropriate, there are many items from recent years. The introduction is properly concluded with the purpose of the work. The description of the results should be removed from the introduction to the topic. The material and methodology are described perfectly, concisely and specifically. I have no major comments. The method of presenting the results and their description is correct. The discussion of the results is extremely interesting, it refers well to the obtained research results. The summary of the work is appropriate.
The manuscript is concise, easy to read, and introduces a new element to microbial science.
In summary, the manuscript is very well written and is relevant to Microorganisms, but requires a few minor corrections.
Minor comments:
Line 23. Please explain the abbreviation ITS.
Line 37. Please change the keywords to something other than the ones in the manuscript title. This will improve the searchability of the article in the database.
Line 72. What do the symbols HQ1-HQ3 and MQ1-MQ3 mean? Please explain.
Line 77-82. Why is the introduction describing the results? This is the summary of the manuscript. Please delete.
Line 148. The abbreviation ANOVA should be explained.
Tables 1 and 2. Please change the individual letters to superscript.
Line 312, 367. Please write Bacillus in italics.
Line 335. Pediococcus should be written in italics.
How do Tables 1 and 2 differ from those in Supplementary Materials S3 and S4?
Reviewer 2 Report
Comments and Suggestions for Authors
The publication presented for review contains a detailed description of starters for a regional product.
It is important to emphasize that, based on the included literature, the subject of the research is analyzed quite intensively from many aspects.
The authors' approach is innovative in some respects, although whether it is important from the point of view of the final product remains a question.
The work is written correctly, in detail and without major reservations in terms of content and logic.
In the last part, the authors suggest the importance of their research, from the point of view of industry, as important indications for industrial composition of the proper aroma of the finished product. Considering the location of Bajjiu and its history, it is difficult not to get the impression that this may be of little importance.
The same question is, who is actually the addressee of the presented publication?
In sum, one cannot make major substantive objections to the authors' work itself, but for the proper substantive tone it would be appropriate to emphasize more strongly what the data provided will find application.
Reviewer 3 Report
Comments and Suggestions for Authors
An interesting manuscript on the effect of temperature preparation on the variability in the microecosystem and physicochemical properties of daqu. There are several issues that need attention:
l. 29. This is a misleading statement. Dominance of Aspergillus is evident only in sample MQ1 (if this color corresponds to Aspergillus and not Wickerhamomyces which is represented by the same color and pattern) and increased population (not exactly dominance) of Saccharomyces is indicated only in MQ2 (Fig. 4).
l. 80, 312. All scientific names should be written in italics.
l. 95. It should read ‘rRNA gene’
paragraph 2.2. Which enzymatic activities represent liquefying and saccharifying activity? A simple reference is not enough, at least provide an in-brief description to enable understanding of the underlying principle. How was fermentation capacity assessed?
l. 174-175, 167-177, 179, 182. Please verify the accuracy of these determinations; such numbers are not indicated by the figures
paragraph 3.3. What is the microbial population of the samples?
l. 306-307. Aspergillus only dominated in 1 sample and the presence of Lactobacillus was not reported.
l. 317. The presence of none of them was reported in the present manuscript.
l. 340-341, 348-349. Which part of this manuscript gives such information? Weissella and lactobacilli were absent, Aspergillus dominated in only 1 sample and Saccharomyces presented higher population that other genera (this is not dominance, a complex microconsortium dominated) in only 1 sample and was almost absent from the rest.
The discussion of the manuscript is not well-prepared and does not provide a solid basis to place the results presented within the relative bibliography. It will benefit if the authors study the review article by Zheng et al. 2011, J. Inst. Brew. 117(1), 82–90.
Reviewer 4 Report
Comments and Suggestions for Authors
This manuscript reports an investigation into the temperature-dependent microbial and physicochemical profiles of Daqu, which critically influence the enzymatic functionality and flavor formation of Baijiu. This study integrated physicochemical assays (moisture, starch, acidity, enzymatic activity) with 16S rRNA and ITS sequencing to analyze HT-Daqu (HQ1–HQ3) and MT-Daqu (MQ1–MQ3) from Sichuan breweries. The findings should elucidate temperature-regulated trade-offs between microbial stress adaptation and metabolic efficiency, providing actionable insights for optimizing Daqu production through targeted microbial management and thermal control to enhance liquor quality. However, the paper is not acceptable in this form and the authors should make some significant modifications. A paper lacking these important features cannot be accepted in “Microorganism”.
Abstract: The abstract has relevant content and good coverage of key aspects of the study (background, methodology, results, implications). However, it is currently too dense and unclear in structure, which may compromise readability, especially for non-specialist readers. It should be revised.
Introduction: The current introduction is too brief and lacks sufficient background to properly frame the study. In particular, there is no clear explanation of what Daqu is, its function in Baijiu production, or how different temperature classifications (HT-, MT-, LT-Daqu) affect microbial and enzymatic dynamics. This omission makes it difficult for readers—especially those unfamiliar with this traditional fermentation system to understand the relevance and implications of the research.
Moreover, the authors include elements of the results within the introduction, which should be avoided. These findings should instead be reserved for the Results or Discussion sections.
The introduction should be comprehensively revised and expanded to:
- Clearly define Daqu and its importance in Baijiu fermentation;
- Define what is Baijiu;
- Outline known differences among HT-, MT-, and LT-Daqu, supported by recent and relevant references;
- Clarify the existing knowledge gap the study aims to fill;
- Avoid inclusion of any experimental results at this stage and set out the objectives of the research.
This restructuring will help provide a solid foundation for the reader and improve the manuscript's accessibility to a broader scientific audience.
In my opinion, the manuscript would benefit from the inclusion of a clear experimental design or workflow scheme. This would help guide the reader through the methodology and improve the overall readability and structure of the article.
Line 85-87: Please provide more detailed information about the Daqu used in the study, including how it was produced, how it was stored prior to analysis, and its basic characteristics at the time of sampling.
Line 190: Remove a point.
Line 111: What are wheat Qu samples?
Table 1: Please ensure that all parameters are presented with the same number of decimal places for consistency.
Table 2: The same as above.
Paragraph 3.5: Why is the distinction between HT- and MT-Daqu no longer maintained in this section? Instead, the term ‘medium-high temperature Daqu’ is introduced without clarifying whether it is a new category or a synthesis of the previous ones. This could be confusing for the reader and should be better clarified.
Conclusion: The authors should briefly discuss the potential practical implications of their findings for the agri-food sector, particularly regarding how microbial or enzymatic differences in Daqu could improve fermentation practices or product quality.
Table S1-S2-S3-S4: Please include statistical analysis (e.g., significance letters, p-values) to allow proper interpretation and comparison of the data.
Round 2
Reviewer 4 Report
Comments and Suggestions for Authors
The authors have thoroughly addressed all previous comments and made significant revisions to enhance the clarity, structure, and scientific rigor of the manuscript. Notably, the introduction has been expanded to provide sufficient background on Daqu and its role in Baijiu fermentation, technical language has been clarified for broader accessibility, and additional experimental details and terminological clarifications have been incorporated. A schematic overview of the experimental design has been added, and statistical analyses have been completed in the supplementary tables as requested. In light of these substantial improvements, the manuscript is now deemed suitable for publication in “Microorganisms”.